# Investigating Genotypic Definitions of Tau Trajectory Subtypes in Alzheimer's Disease using Tree-Based Models

**The Alzheimer's Disease Neuroimaging  Institute**[*]

## Abstract

As life expectancies increase, dementia and neurodegenerative diseases affect increasing portions of the population. Alzheimer's disease (AD) in particular has shown increasing variation in disease progression which suggests subtyping as an avenue for improving understanding and treatment of the condition. Recent research in neuroimaging has revealed distinct phenotypes of AD but a genetic explanation of these subtypes has yet to be uncovered. To that end we attempt to classify these subtypes from SNPs. Multiple tree-based models demonstrate that we found no correlation between tau trajectories and clinically SNPs in a small sample of ADNI patients. Beyond the possible confounds of small sample size, our results are also consistent with the lack of previous studies that implicate SNPs in tau pathology. Based on our discoveries, we include multiple suggestions for future studies into tau-disposition of AD as well as suggestions for linking SNP data to other kinds of phenotypic subtypes of AD.

## 1   Introduction

Human disease has long been notoriously difficult to study as diagnostics are first defined by a broad swath of symptoms. While symptoms are necessary to determine what may ail a patient, such a broadly phenotypic approach can result in multiple conditions with distinct etiologies receiving the same diagnosis. This complexity obfuscates efforts to translate observed symptoms to their underlying causes (Zanin et al., 2018). In order to develop effective treatments, it is necessary to further our understanding of the biological disease of interest. Recent advances in computational biology have sought to increase our understanding of disease at a genetic level. These advances have been driven by the increasingly accepted belief that a set of private genetic features can lead to distinct clinical manifestations in disease. As such, further analysis into genetic heterogeneity is critical when making sense of disease pathology (McClellan & King, 2010).

Substantial progress has already been made in oncology, especially in breast cancers where subtypes can be classified according to differing gene expression: luminal A, luminal B, Her2-enriched, and basal-like. The distinct genetic alterations that characterize these subtypes subsequently result in distinct clinical prognosis for breast cancer patients, and raise the possibility for personalized healthcare (Banerji et al., 2012). This effort to capture heterogeneity in cancers has now resulted in over a hundred types of cancers that are used in the diagnostic process. In fact, a diagnosis of cancer now requires a patient to receive additional testing in order to specify the precise subtype and stage of a cancer.

---

[*]*Data used in preparation of this article were obtained from the Alzheimer's Disease Neuroimaging Initiative (ADNI) database (adni.loni.usc.edu). As such, the investigators within the ADNI contributed to the design and implementation of ADNI and/or provided data but did not participate in analysis or writing of this report. A complete listing of ADNI investigators can be found at: `http://adni.loni.usc.edu/wp-content/uploads/how_to_apply/ADNI_Acknowledgement_List.pdf`

Preprint. Under review.

However, within the Alzheimer's disease (AD) research community and the pharmaceutical industry, Alzheimer's is treated as a single, monolithic disease (Au et al., 2015; Devi & Scheltens, 2018). A paradigm similar to that in cancer may very well be justified in attempting to characterize AD into subtypes with distinct clinical manifestations especially when considering the current genetic understanding of both diseases. In particular, both cancer and late onset AD are caused by a somatic mutations which accumulate over the course of a patient's life. A patient's accumulated somatic mutations may ultimately lead to distinct clinical manifestations of the disease. Thus, we posit that the path to the development of effective treatments for AD requires a similar paradigm to that of cancer in which we must first identify distinct subtypes of AD in order to better unravel the mysteries of AD.

## 1.1 Heterogeneity in Alzheimer's Disease

The study of heterogeneity in Alzheimer's disease is not an entirely novel idea. There has been a substantial pool of work dedicated to capturing this heterogeneity through a combination of neuroimaging techniques such as structural MRI scans and PET scans as well as a set of neuropsychiatry profiling in memory, visuospatial functioning, and language (Habes et al., 2020; Vogel et al., 2021). However, relatively little is known about the pathophysiological mechanisms that underlie AD development and the distinct clinical presentations of AD, especially within the context of gene expression (Habes et al., 2020). What little is known is that germline mutations in *APP, PSEN1, PSEN2* genes significantly increase risk for early-onset AD and follows a Mendelian inheritance pattern while somatic mutations in *APOE4* corresponds to an increased risk for late-onset AD (Miller et al., 2021; Freudenburg-Hua et al., 2018). In fact, it has been estimated that genetic factors contribute to 58-79% of an etiologic role in late-onset AD (Gatz et al., 2006). Of these variants, work into the analysis of single nucleotide variants (SNVs) and single nucleotide polymorphisms (SNPs) may show the most promise due to their utility in subtyping cancer (Zhang et al., 2020). Taking inspiration from this line of thought, we elect to study SNPs involved in AD. But with only 30.6% of genetic variance explained by known AD single-nucleotide polymorphisms (SNPs), there is a need to further expand the scope of SNP studies in order to fully understand the nature of AD (Freudenberg-Hua et al., 2018).

## 1.2 Expanding Genetic Analysis of AD Beyond the Realm of Exome Sequencing

Our study takes advantage of two major developments in the world of computational biology. First, imaging genomics is an emerging field that connects neuroimaging techniques with genetic information. Over the last few years, imaging genomics has made substantial progress in large part due to the increased accessibility of brain scans from patients, making it possible to potentially provide causal, or at the very least correlatory evidence between genotypic differences and phenotypic differences in the brain. In this study, the use of neuroimaging techniques is particularly useful in highlighting the effects of genetic changes as the phenotypic differences observed are are much closer to underlying biology than standard behavioral or diagnostic measures which are commonly used in studying neurodegenerative diseases (Thompson et al., 2010).

Second, over the last few years, the cost of emerging sequencing technologies, particularly, whole genome sequencing (WGS) and whole exome sequencing (WES) have decreased significantly, allowing for the possibility of further extensive study. These sequencing technologies can better capture such rare variants which are not in linkage disequilibrium with SNPs on a genotyping array used for GWAS. Overall, they are superior when it comes to determining genotypes of rare variants with high accuracy (Hogland et al., 2019). In our study, we aim to take advantage of the scope and accuracy of WGS in order to detect rare variants that may play a role in distinct manifestations of Alzheimer's. More crucially, we choose to include mutations that exist within non-coding regions of the genome as the exome comprises 1-2% of the human genome, making it likely that patients may harbor relevant mutations in regions beyond the exome. Additionally, there has been little work into studying non-coding variants that may affect AD due to the technical limitations in Genome Wide Association Studies (GWAS) even though previous literature does highlight the potential role in non-coding genomic information in AD development (Zhou et al., 2021). The crux for this work is therefore motivated by our hypothesis that the lack of analysis into these non-coding variants is one of the major reasons as to why we have yet to conclusively uncover distinct subtypes.

## 2 Methods

### 2.1 Image Acquisition and Preprocessing

Image data from 211 patients alongside their standardized update update value ratio (SUVR) scores used in our analysis was obtained from the Alzheimer's Disease Neuroimaging Initiative (ADNI) database (adni.loni.usc.edu). The ADNI was launched in 2003 as a public-private partnership. The primary goal of ADNI has been to test whether serial magnetic resonance imaging (MRI), positron emission tomography (PET), other biological markers, and clinical and neuropsychological assessment can be combined to measure the progression of mild cognitive impairment (MCI) and early Alzheimer's disease (AD). For up-to-date information, see www.adni-info.org. We obtained brain parcellations of the brain using the Desikan-Killiany (freesurfer) atlas and used these to obtain mean SUVR values within various regions of interest (Klein & Tourville, 2012). We selected five regions of interest in both the left and right hemisphere: parietal, frontal, occipital, temporal, and medial temporal lobe (MTL), resulting in ten spatial features. These regions were selected according to suggestions from previous work in which variation in AD pathology occurs within these specific lobes (Ossenkoppele et al., 2015; Vogel et al., 2021). Every patient's SUVR scores were subsequently compiled into distributions for each ROI. Using Vogel et al., 2021's procedure, we use a two-component Gaussian mixture model to define a normal and abnormal distribution. This model was subsequently used to create z-scores by normalizing SUVR values from the mean of the normal distribution. Using pySuStaIn, an algorithm which infers disease progression patterns and severity from cross-sectional data, z-scores along with arbitrary severity scores of $z = 2, 5, 10$ are used to generate 5 subtype labels based on distinct phenotypic trajectories of tau depositions in the brain (Aksman, et al., 2021). The labels generated from the image data classified 135 of the 211 patients as part of the control group (no Alzheimer's). For the remaining patients, 30 were classified as "subtype 1", 17 were classified as "subtype 2", 19 were classified as "subtype 3", and 10 were classified as "subtype 4". As can be clearly seen, the subtypes associated with each label has an imbalance which will be addressed during the validation step by ensuring that the validation sets possess a similar distribution to that of the training set.

### 2.2 Genetic Data Acquisition and Pre-Processing

WGS data was obtained from the ADNI database for the 211 patients whose tau-PET images were available. Considering the small sample size of our data, we opt towards the use of SNPs instead of SNVs to prevent overfitting. SNPs of unknown origin were removed and the remaining SNPs were filtered into two sets of features containing clinically significant SNPs from either non-coding regions or coding regions according to the Single Nucleotide Polymorphism Database (dbSNP). Coding SNPs are of particular importance in this study for three reasons. First, AD related mutations discovered thus far rest primarily in coding regions. Second, the connection of genotype to phenotype coding SNPs can produce known phenotypic changes at the protein-level. Third, coding regions are highly conserved due to selective pressure which additionally makes coding SNPs critical when determining genetic links to disease. Through such an analysis on coding SNPs, we can better capture the efficacy of our model by incorporating domain knowledge and thus biological meaning to our results. However, given that previous AD research has not been able to elucidate subtypes of AD, we further analyze the effect of SNPs from non-coding regions on these phenotypic differences. To effectively do so, coding SNPs were removed to reduce the bias they naturally introduce into WGS analyses.

### 2.3 Model

For this work, we use two explainable and popular machine learning models to capture genetic heterogeneity in AD. A particular emphasis on interpretability is made to ensure that other computational biologists are able to better elucidate the underlying mechanisms that drive AD subtyping through our work.

We first train a decision tree using two distinct sets of features: non-coding and coding SNPs. A decision tree model has traditionally higher interpretability when compared with other machine learning models. The ability to identify which features the model values provides valuable insight into the specific SNPs that form the genetic basis for the phenotypic subtypes of AD. We further extend the decision-tree model using XGBoost, a regularizing gradient boosting framework, we retain

the interpretability advantages of standard decision trees while also having faster execution speeds and potentially better performance on many machine learning problems.

We then train a random forest model in capturing genetic heterogeneity of AD using the Python package scikit-learn. Much like the previous two models, we used non-coding and coding SNPs as distinct sets of features to use for our model. A random forest classifier is used as they often reduce over fitting and boost model performance through the training of multiple decision trees on different subsets of features. This, however, may come at the cost of interpretability relative to a decision-tree model but is still more interpretable than other alternative machine learning models with similar performance, such as neural networks.

## 2.4 Evaluation

During our evaluation step of our model, we perform two types of validation. First, in order to ensure that the training and validation sets were representative of the distribution of patient subtypes in the original dataset, we generated a stratified split of the dataset after randomizing the order of the patients, with eighty percent of patients (168 patients) in the training dataset and twenty percent of patients (43 patients) in the validation dataset. Afterwards, we computed the training accuracy and computed the accuracy and precision on the validation set to measure model performance.

Second, we performed five fold cross validation on the models using scikit-learn's cross_val_score to gain a better understanding of how the models would behave on independent datasets. In doing so, we provide a greater understanding of how our model might perform in a real-world context. We used the distribution of validation accuracies across the five folds as measures of model performance.

## 3 Results

### 3.1 Performance of Decision Tree Models

We used the decision tree classifier from the scikit-learn package. We first performed parameter tuning by computing the validation accuracy for trees with depth limitation ranging from 2 to 10 and selected the one which on average had the best validation accuracy. For non-coding SNPs the best performing model has a depth limit of 3 for the complete decision tree while the best decision tree model for coding SNPs has a depth limit of 5. We retained the default values for all other parameters. Using these parameters, we computed accuracy, precision, and confusion matrices on the stratified training and validation datasets (see Figure 1, 8), and computed accuracy from 5-fold cross validation(see Figure 2). Complete decision trees can be found in Figure 15 and Figure 16 in the Appendix.

| Dataset | Training Accuracy | Validation Accuracy | Validation Precision |
|---|---|---|---|
| Noncoding SNPs | 0.721 | 0.627 | 0.55 |
| Coding SNPs | 0.785 | 0.651 | 0.57 |

Figure 1: Performance of Decision Tree Model on Coding and Noncoding SNPs with Stratified Training and Validation Set

| Dataset | Fold 1 | Fold 2 | Fold 3 | Fold 4 | Fold 5 | Mean | Standard Deviation |
|---|---|---|---|---|---|---|---|
| Noncoding SNPs | 0.534 | 0.452 | 0.571 | 0.476 | 0.523 | 0.511 | 0.047 |
| Coding SNPs | 0.465 | 0.571 | 0.476 | 0.667 | 0.547 | 0.545 | 0.082 |

Figure 2: Cross Validation Accuracies of Decision Tree Model on Coding and Noncoding SNPs

After observing the performance of the decision tree model, we decided to investigate if using a typically higher performing variant of the decision tree would have better classification performance. As a result, we then trained the XGBoost decision tree classifier on the training set with 168 patients using coding and non-coding SNPs as the features. Since XGBoost only works with binary labels,

| Dataset | Training Accuracy | Validation Accuracy | Validation Precision |
|---|---|---|---|
| Noncoding SNPs | 1.000 | 0.621 | 0.47 |
| Coding SNPs | 1.000 | 0.637 | 0.51 |

Figure 3: Performance of XGBoost Decision Tree Model on Coding and Noncoding SNPs with Stratified Training and Validation Set

| Dataset | Fold 1 | Fold 2 | Fold 3 | Fold 4 | Fold 5 | Mean | Standard Deviation |
|---|---|---|---|---|---|---|---|
| Noncoding SNPs | 0.613 | 0.628 | 0.613 | 0.598 | 0.628 | 0.616 | 0.011225 |
| Coding SNPs | 0.628 | 0.643 | 0.613 | 0.613 | 0.628 | 0.625 | 0.011225 |

Figure 4: Cross Validation Accuracies of XGBoost Decision Tree Model on Coding and Noncoding SNPs

we used a multi-label-classifier on top of the XGBoost model to predict on multi-labels. For the XGBoost model, we set gamma to 0.1, and the objective to binary logistic. Its performance is detailed in Figures 3 and 4 and the confusion matrices are presented in 9, 13.

## 3.2 Performance of Random Forest Models

Since the results of the decision tree and XGBoost models were comparable, we wanted to investigate whether a more stable tree based model would yield better predictions. We elected to train a random forest model for this task. Figure 5 shows the performance for the random forest model using coding and non-coding snps as features on the stratified training and validation datasets. Figure 6 shows the performance of the random forest model in cross validation. From the results, It should be noted that for both coding and non-coding SNPs used as training features, the test set results are identical, because the random forest model at hand seemed to be unable to learn a better classification scheme than simply classifying every patient in the validation set with the most common label (control group) as seen from its confusion matrix (see 10 and 14).

| Dataset | Training Accuracy | Validation Accuracy | Validation Precision |
|---|---|---|---|
| Noncoding SNPs | 1.0 | 0.651 | 0.424 |
| Coding SNPs | 1.0 | 0.651 | 0.424 |

Figure 5: Performance of Random Forest Model on Coding and Noncoding SNPs with Stratified Training and Validation Set

| Dataset | Fold 1 | Fold 2 | Fold 3 | Fold 4 | Fold 5 | Mean | Standard Deviation |
|---|---|---|---|---|---|---|---|
| Noncoding SNPs | 0.628 | 0.628 | 0.643 | 0.643 | 0.659 | 0.640 | 0.011 |
| Coding SNPs | 0.628 | 0.628 | 0.643 | 0.643 | 0.659 | 0.640 | 0.011 |

Figure 6: Cross Validation Accuracies of Random Forest Model on Coding and Noncoding SNPs

## 3.3 Comparative Performance of Models

Figure 7 compares the performance of the decision tree model, the XGBoost model, and the random forest model on the noncoding SNP dataset. Similarly, Figure 11 compares the performance of the three models on coding SNPs. The decision tree model seems to perform better on the validation set than the XGBoost model and the random forest model. However, the difference is not very significant. Observing what errors are made by the models through confusion matrices offers insight

into their comparative performances. Figures 8 and 12 show that the decision tree is also almost always predicting a label of zero. Figures 9 and 13 show that the XGBoost model mostly predicts zero labels like the random forest and the decision tree. In Figures 10 and 14, the confusion matrices for the random forest show that the model always predicts a label of zero (no AD). This suggests that the performance of the decision tree models on this dataset are unstable since the random forest aggregates the results of a large number of decision trees. This could potentially be a result of the severe class imbalances in the dataset with over sixty percent of the patients being in the control group and certain classes of Alzheimer's patients having very few patients (subtype 4 has ten patients). The limited size of the dataset is another potential cause for this behavior since the high training accuracies of all three models suggests that overfitting me be an issue. Another significant observation is that validation performance seems to be similar on the coding SNP dataset and the noncoding SNP dataset across all models. This suggests that the models are not learning the relationships between the SNPs and the tau trajectories or that there is not a significant relationship between the SNPs and the tau trajectories based labels.

| Model | Train Accuracy | Validation Accuracy | Validation Precision | Mean Cross Validation Accuracy |
|---|---|---|---|---|
| Decision Tree | 0.721 | 0.627 | 0.55 | 0.511 |
| XGBoost | 1.000 | 0.621 | 0.47 | 0.616 |
| Random Forest | 1.000 | 0.651 | 0.42 | 0.640 |

Figure 7: Comparing Performance of Models on Noncoding SNPs

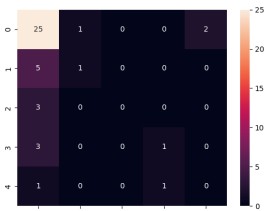

Figure 8: Heatmap of Decision Tree Confusion Matrix on Noncoding SNPs

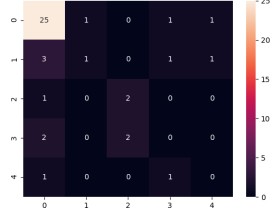

Figure 9: Heatmap of XGBoost Decision Tree Confusion Matrix on Noncoding SNPs

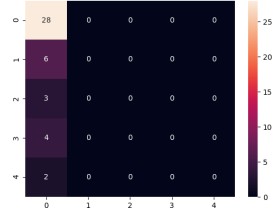

Figure 10: Heatmap of Random Forest Confusion Matrix on Noncoding SNPs

| Model | Train Accuracy | Validation Accuracy | Validation Precision | Mean Cross Validation Accuracy |
|---|---|---|---|---|
| Decision Tree | 0.785 | 0.651 | 0.57 | 0.545 |
| XGBoost | 1.000 | 0.637 | 0.51 | 0.625 |
| Random Forest | 1.000 | 0.651 | 0.424 | 0.640 |

Figure 11: Comparing Performance of Models on Coding SNPs

## 4 Discussion

In this exploration, we attempted to find a SNP-level explanation for phenotypic subtypes observed in AD. First, we curated the SNP features to in the feature space to prevent over-fitting and additionally leverage domain knowledge. Limiting our feature space to non-coding mutations has previously proven successful in somatic disease subtyping. Unfortunately, this approach did not produce high levels of accuracy with the phenotypic labels that were generated. Addition of coding variants also

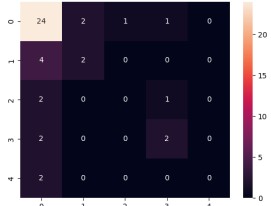 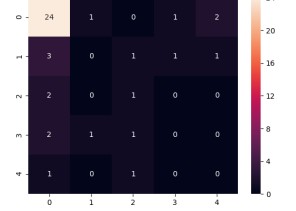 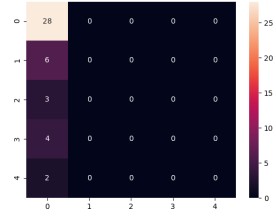

Figure 12: Heatmap of Decision Tree Confusion Matrix on Coding SNPs

Figure 13: Heatmap of XGBoost Decision Tree Confusion Matrix on Coding SNPs

Figure 14: Heatmap of Random Forest Confusion Matrix on Coding SNPs

did not result in higher accuracy. An analysis of coding SNPs alone suggests that the lack of efficacy of our predictive models may be due to small sample sizes.

Future efforts should include larger amounts of patients to offset the high variance observed in SNPs. Additionally, larger sample sizes would improve the stability and accuracy of imaging labels in the first place, as larger feature spaces alongside larger datasets have yielded better results for other researchers. Unfortunately, given data privacy concerns, running an analysis with these data would be better sorted to independent research efforts where data usage is not limited to public datasets. The inclusion of other datasets also helps to add a variety of clinical patients. Another benefit that may arise are more diverse populations that can help minimize population genetic effects that may occur in future studies.

While SNP and SNV based approaches may allow for more nuance than GWAS in genetic analyses of AD, it is worth noting that most relevant genes and loci of interest identified by previous GWAS efforts are not involved in Tau-pathology. Indeed recent research efforts have suggested that the kinds of phenotypic subtypes revealed by Vogel et al from neuroimaging data might be better elucidated by gene expression data (Zheng and Xu 2021). So while it may be possible that non-coding SNPs and rare variants can add some insight into AD subtyping, subtypes from imaging data in AD may be better explained by transcriptomic datasets. In searching for an explanation for these gene expression changes, epigenomic and chromatin data might also prove more useful.

SNP data may be capturing a different aspect of AD and may still be useful when used in conjunction with other forms of phenotypic data. Large-scale studies of phenotypic data have are already of particular interest to clinicians, LASI-DAD being a particular example of the kinds of data that may be more plentiful to enable the kinds of scales that may be necessary for WGS analyses to be able to tease out subtypes. As another possible suggestion for individuals interested in WGS, UKBioBank has a variety of SNP and phenotypic data that may be leveraged for connecting phenotypes and genotypes in AD. As already mentioned, SNPs are subject to population genetic effects which might enable a more epidemiological approach. Finally, when considering data modalities SNP data may be more useful at the single-cell scale. It is easier to separate metastatic cells from health cells in cancer compared to AD affected cells. A wide variety of cells are impacted by AD and somatic mutations exhibit a degree of mosaicism that may necessitate a finer grained analysis.

In an aging world, neurodegenerative diseases are becoming an increasing health concern. With increased presences of diseases like AD, varying pathologies emerge. Subtyping is a natural effort to make sense of the difference in disease progression. Imaging data in AD suggests a phenotypic explanation for distinct sets of symptoms. To enable better treatments and patient outcomes, a biological explanation of AD at a genetic level remains a prominent goal in AD research. Current treatments for AD attempt to ameliorate symptoms, but once the global research community is armed with the etiologies of AD, we may soon be able to improve these treatments and address the problem at its source.

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

# A  Appendix

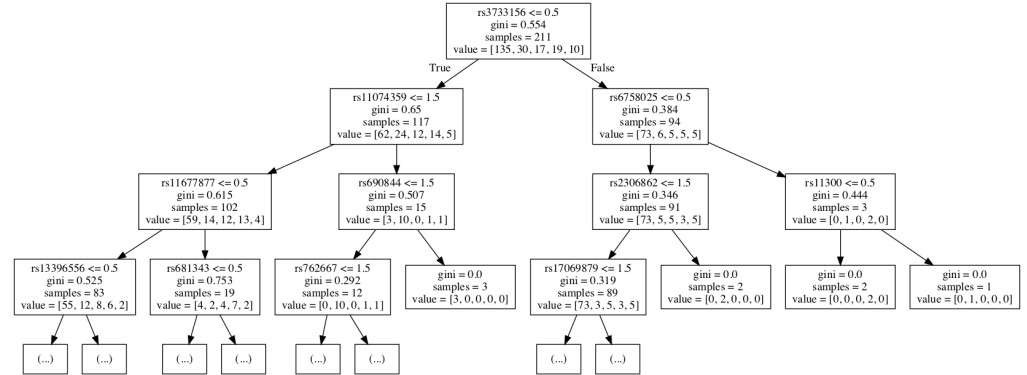

Figure 15: Decision Tree with patient labels from pySuStaIn with depth limit 3 based on non-coding region

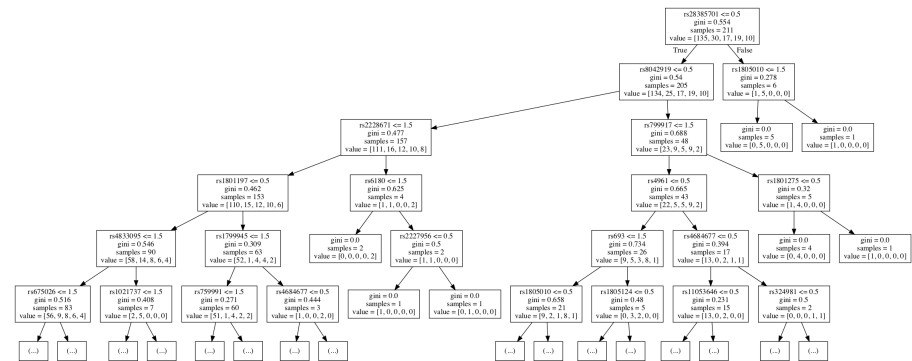

Figure 16: Decision Tree with patient labels from pySuStaIn with depth limit 5 based on coding region

