# OpenReview forum: "Investigating Genotypic Definitions of Tau Trajectory Subtypes in Alzheimer’s Disease using Tree-Based Models"
_uoft.ai/University_of_Toronto/2021/ProjectX — Submitted to ProjectX2021_

### Official Review · Reviewer_jjxk · 2022-02-07
**Investigating Genotypic Definitions of Tau Trajectory Subtypes in Alzheimer’s Disease using Tree-Based Models**

**Rating:** 5
**Confidence:** 4

**Review:**

**Connection to Current Science (science and practice)**

1

- The approach is well framed in Section 1.
- Shapley values, information gain, or simple statistical filtering would be a preferable approach to using a single decision tree, for explainability
- this paper does not use much in the way of cutting-edge technology, either from the machine learning or the explainability perspectives.
- the discussion is somewhat superficial (e.g., it would have been good to have more data).
- it is suggested that biological explanations are important to enable better treatment, but the exact line between aetiology and treatment is not drawn (albeit out of scope, perhaps), and not very much in the way of explanation is given.


**Clarity of Communication**

2

- generally, the structure is logical, and each section is concise. The writing is clear with a few mistakes (e.g., “better sorted” instead of “better suited”).
- Figures 1 and 3 should not include training accuracy. Figures 2 and 4 can merely report the last two columns.
- None of Figures 1-4 should be captioned as Figures, but as Tables.
- Figures 8-10 and 12-14 are hard to read without zooming in, and are fairly uninformative.


**Methodological Quality**

2

- it would have been preferable to test for normality in Section 1 of each subgroup in the GMM, but this is fine.
- the arbitrariness of the severity scores is somewhat questionable and should be better explained.
- some unsubstantiated assumptions are made in Sec 2.3 (e.g., about performance relative to neural networks)

**Reproducibility**

0.8

- The description of preprocessing in Section 2.2 is appreciated.
- the exact nature of the data used for training is not specified sufficiently (e.g., to what do the various rs.* variables in Figures 15 and 16 refer?).

---

### Official Review · Reviewer_8XPQ · 2022-02-14
**Great clinical concept but methods could use some work**

**Rating:** 6
**Confidence:** 3

**Review:**

1) Connection to Current Science (science at6nd practice) [SCALE 0(low) – 3(high)]
•	Does this work show knowledge of the existing state of the field?
•	Does this work add something new to the literature?
•	Do the teams discuss what their pathway to implementation looks like?
Comments: Great topic. Focus is clearly stated. Impact is quite important, and need is demonstrated. 3/3

2) Clarity of Communication [SCALE 0(low) – 2(high)]
•	Is the writing clear and easy to follow? yes
•	Are data visualization techniques chosen and labelled well? I would have chosen a graphical way instead of tables to compare accuracy and precision
•	Is there a clear logical structure to the paper? yes
1.5/2

3) Methodological Quality [SCALE 0 (overly complicated models that are ill-suited to the problem) - 4 (develops new methods that provide insight into an important problem)]
•	Is this paper making reasonable assumptions?
•	Does this paper use methods that are appropriate to the problem at hand?
•	Does the paper introduce new and interesting methods? (old methods applied well is good as well)
•	Does the paper avoid needless model complexity?
Comments: Why was precision and accuracy evaluated? How does that translate to clinical metrics? Mention of positive predictive power and specificity would have been nice. Recall was excluded but is an important metric that should be evaluated between models.
Sample size is small. This is a big limitation for application of machine learning tools… Also no test set was included. Metrics from the validation set are not that informative compared to test set as often times these metrics fall significantly. This is quite a limitation for this study. Always train hyperparameters on the validation test and assess the model itself through a test set.
1.5/4

4) Reproducibility [SCALE 0(low) to 1(open source code)]
•	Does this paper seem reasonable as work conducted in a span of 5 months? yes
•	Has the team been open for others to reproduce the paper’s results? No open source code available
0.5/1


Total = 6.5/10

---

### Official Review · Reviewer_W2jh · 2022-02-15
**The authors propose machine learning tree-based approaches to investigate whether coding and non-coding variants can aid in identifying Alzheimer’s Disease subtypes defined using neuroimaging technologies.**

**Rating:** 5
**Confidence:** 4

**Review:**

Overall, I find the topic of interests and relevant in the field and commend the authors for their work. The manuscript is well written, organized and follows a clear structure. Nonetheless, I have several comments regarding the quality, clarity, and originality of this work.

**Major comments**

For the most part, the authors appear to have applied the tree-based approaches ‘as-is’ or ‘out-of-the-box’. That is, despite some hyperparameter tuning was performed, little to no effort was made to go beyond preparing a dataset and plug it into the scikit-learn functions. This is my main criticism of this work.

Related the point above, additional variant prioritization could have been achieved using biological information, e.g., functional annotations. Further, although the rationale for looking at non-coding variants separately is well justified, I wonder if looking other variant categorizations, e.g., synonymous, non-synonymous, etc., would have been helpful.

Although the authors seem to have curated the list of SNPs that were used as input to the models, overfitting may still be a problem as hinted by the 100% train accuracy displayed by the XGBoost and Random Forest approaches. This may not be too surprising because of the large number of SNPs that were still considered. Moreover, I find the methods section lacks enough details on the number of features that were considered in each of the evaluated models, i.e. how were the curated lists generated?

As pointed out by the authors, severe class imbalances composed with small sample sizes may explain (at least partially) the poor performance. On this note, oversampling strategies such as SMOTE could have been used. In addition, given the amount of data and these severe imbalances, are the 5 subtypes sufficiently justified?

**Minor comments**

It would have been helpful if the authors had included heir definitions of accuracy and precision as part of the methods section.


**Pros**

The authors elaborate on several interesting future areas of research considering their negative results.

**Cons**

No code was made available to reproduce the results.
The evaluation metrics did not consider any uncertainty associated to them; repeated stratified splitting and repeated cross-validation could have been attempted to circumvent this issue.

---

### Official Review · Reviewer_tTNP · 2022-02-16
**The paper tackled an interesting problem of identifying imaging classes of AD and relating them to germline genetics. Unfortunately, the authors appear to have little knowledge of human genetics (esp in AD), the data are likely too limited, and the methodology is not suitable for the problem at hand, nor is the attempt successful.**

**Rating:** 4
**Confidence:** 5

**Review:**

The authors aim to define AD classes based on neuroimaging features and then build classifiers from germline SNPs for those classes.
While this is an important problem, I am afraid that the authors make a lot of sweeping statements and assumptions that are either unsupported by current knowledge of AD genetics or simply wrong. For example, AD genetics is not primarily driven by somatic mutations and the architecture is very different than that in cancer. Variants are mostly in non-coding regions (exome sequencing does not preclude that b/c variants are imputed based on LD...). There are too many such statements to count.

Next, the authors take several unclear steps: they avoid any genetic association approach -- the appropriate way to make causal inference in genetics, and instead take a classification approach. It remains entirely unclear why they featurized the imaging data in the way they did, and how they use the SNP features. Ultimately (and sadly, not surprisingly) the classifiers are not successful.

Overall, while the work is an interesting start of a direction, the actual application is not successful.

---

### Decision · Program_Chairs · 2022-02-19

NA